# Ionic and Non-Ionic Counterparts Based on Bis(Uracilyl)Alkane Moiety with Highest Selectivity Towards Acetylcholinesterase for Protection Against Organophosphate Poisoning and Treating Alzheimer’s Disease

**DOI:** 10.3390/ijms26083759

**Published:** 2025-04-16

**Authors:** Irina V. Zueva, Liliya F. Saifina, Liliya M. Gubaidullina, Marina M. Shulaeva, Alexandra D. Kharlamova, Oksana A. Lenina, Grigory P. Belyaev, Albina Y. Ziganshina, Shan Gao, Wenjian Tang, Vyacheslav E. Semenov, Konstantin A. Petrov

**Affiliations:** 1Arbuzov Institute of Organic and Physical Chemistry, FRC Kazan Scientific Center of RAS Arbuzov str., 8, Kazan 420088, Russiadimple@mail.ru (L.F.S.); gubaidullina@iopc.ru (L.M.G.); mshulaeva@iopc.ru (M.M.S.); niksashenka@yandex.ru (A.D.K.); leninaox@mail.ru (O.A.L.); gregoir4@gmail.com (G.P.B.); az@iopc.ru (A.Y.Z.); kpetrov2005@mail.ru (K.A.P.); 2School of Pharmacy, Anhui Medical University, Hefei 230032, China; gs129600@outlook.com; 3Graduate School of Biology, Institute of Fundamental Medicine and Biology, Kazan Federal University, 18 Kremlyovskaya str, Kazan 420008, Russia

**Keywords:** acetylcholinesterase, butyrylcholinesterase, 6-methyluracil derivatives, organophosphate poisoning, peripheral anionic site, Alzheimer’s disease

## Abstract

A series of bisuracils, in which uracil and 3,6-dimethyluracil moieties were bridged with a polymethylene spacer, and the uracil moiety contained a pentamethylene radical with ionic and non-ionic aminobenzyl groups, were synthesised. These bisuracils have been identified as cholinesterase inhibitors with exceptional selectivity for acetylcholinesterase (AChE) over butyrylcholinesterase (BuChE). These bisuracils, which have been identified as highly effective AChE inhibitors, demonstrated activity at nano- and sub-nanomolar concentrations, with exceptional selectivity for AChE over BuChE. In kinetic studies of lead bisuracils **2b** and **3c**, both compounds exhibited mixed-type inhibition against AChE and BuChE. Additionally, molecular dynamic simulations demonstrated robust and stable interactions of **2b** and **3c** with the binding sites of their target. Bisuracil **2b** showed significant potential for protection of AChE from irreversible inhibition by paraoxon; the most effective dose of 0.01 mg/kg was shown to reduce mortality in paraoxon-poisoned mice. Bisuracil **3c** effectively inhibited brain AChE activity, reversing scopolamine-induced amnesia in mice at a dose of 5 mg/kg, which indicates its potential for cognitive enhancement. These findings position ionic bisuracils as promising prophylactics against organophosphate poisoning and non-ionic bisuracils as viable candidates for Alzheimer’s disease therapeutics.

## 1. Introduction

Cholinesterase inhibitors represent a class of compounds that interfere with the action of the enzyme cholinesterase to prevent the breakdown of acetylcholine, which is an important neurotransmitter in the nervous system. Such compounds, which interact with the enzyme as a primary target, can be used both as drugs and toxins [1]. The nature of the interaction between the inhibitor and the enzyme determines whether the inhibitor will be reversible or irreversible. Reversible inhibitors are widely used in the treatment of neurodegenerative disorders. Particular attention has been paid to currently approved cholinesterase inhibitors donepezil, rivastigmine, and galantamine in the pharmacotherapy of Alzheimer’s disease (AD) [2]. Reversible cholinesterase inhibitors competitively bind to AChE, preventing irreversible inhibition by OPs. This protective effect is most effective before OP exposure, as it temporarily shields the enzyme’s active site [3,4]. However, once OP-induced ‘aging’ occurs (irreversible phosphorylation of AChE), reversible inhibitors are ineffective, highlighting the need for prophylactic use. Reversible cholinesterase inhibitors are additionally utilised as pre-treatment for poisoning by irreversible cholinesterase inhibitors, represented by organophosphorus compounds (OPs), which are widely used as pesticides [5].

OPs have been identified as a prevalent cause of fatal poisonings, especially in rural areas of developing countries [4,6,7]. The prevailing standard of care for acute OP poisoning, which includes the administration of atropine, a muscarinic ACh receptor blocker, as well as various oxime compounds for activating acetylcholinesterase (AChE), has not met patient expectations [8]. However, if exposure to OPs is anticipated, preventive measures include the use of non-OP AChE inhibitors, which are capable of reversibly binding to the enzyme, thereby protecting its active site from irreversible inhibition by OPs [5,9].

Representing the most common cause of adult-onset dementia in people over 65, AD is characterised by progressive impairment of learning and memory functions at an early stage and ultimately resulting in death as the disease progresses [10]. The most extensively studied pathological features of AD are β-amyloid (Aβ) plaques primarily composed of aggregated forms of the Aβ peptide and neurofibrillary tangles, characterised by the presence of hyperphosphorylated tau protein associated with microtubules [11]. Current symptomatic treatments for AD include cholinesterase inhibitors (e.g., donepezil) that align with the cholinergic hypothesis, though disease-modifying therapies targeting amyloid-β and tau are under investigation. While current symptomatic treatments for AD primarily target the cholinergic hypothesis—which links cognitive decline to acetylcholine deficiency and loss of cholinergic neurons [12]—emerging evidence highlights the multifaceted role of AChE in neurodegeneration. Beyond its catalytic function, AChE accelerates amyloid-β aggregation [13] and interacts with tau pathology [14,15], suggesting broader involvement in AD progression. Notably, non-catalytic roles of AChE in neuroinflammation and synaptic plasticity [16] further complicate therapeutic targeting. By partially inhibiting these enzymes, acetylcholine levels can be increased, leading to improvements in learning and memory [17,18,19].

In a recent work, we reported a new class of selective AChE inhibitors with a bisuracil scaffold. Based on the proposed concept, bisuracils described by formula **1** were successfully designed and synthesised (Figure 1). These compounds consisted of 3,6-dimethyluracil and uracil rings bridged to each other by a polymethylene spacer, with the N1 atom of the uracil ring containing a pentyl(diethylammonium)benzyl radical. Electron-withdrawing nitro- and trifluoromethyl substituents were located in the o-position of the benzene ring of the benzyl moiety. The efficacy of these ionic bisuracils **1** as inhibitors of AChE was demonstrated at nano- and sub-nanomolar concentrations, exhibiting a highest degree of selectivity for AChE over BuChE. The lead compound in this series, namely bisuracil **1** with a hexamethylene spacer between uracil moieties and the trifluoromethyl substituent at the benzene ring, has been demonstrated to enhance the efficacy of antidotal therapy as a pre-treatment for poisoning by OPs [20].

The present study expands the concept of selective AChE inhibitors based on the bisuracil scaffold in two directions, the first consisting in the introduction of nitrile substituents into bisuracils—in particular, bisuracil **2** (Figure 2). In this instance, it is particularly important to consider the extent to which the substitution with electron-withdrawing nitrile groups in compounds with a similar structure to bisuracil **1**, with electron-withdrawing nitro- and trifluoromethyl groups, could affect their activity against AChE. It has been demonstrated that nitrile groups exhibit a specific effect on biological activity [21,22]. In the present study, a series of bisuracil **2** was afforded, which exhibited the same structure as bisuracil 1, but, in contrast, carried nitrile substituents at the benzene ring. Earlier experience in studying the effect of the nitrile group on the activity of AChE inhibitors using 1,3-bis[ω-(substituted benzylethylamino)alkyl]uracils served as an example. The hypothesised nitrile-group effect indeed manifested in an outstanding affinity and selectivity against AChE exhibited by the studied compounds [23].

In the second direction of expanding the concept of bisuracil scaffold-based AChE inhibitors, we aimed to design and synthesise bisuracils that are isostructural to bisuracil **2** but have a non-ionic rather than a charged structure. Although the neutral bisuracil **3** (Figure 2) is described as having the same bisuracil scaffold as compounds **1** and **2**, it contains a non-ionic pentyl(ethylamino)benzyl radical at the uracil ring. It is known that charged AChE inhibitors cannot cross the blood–brain barrier (BBB), whereas non-ionic AChE inhibitors are able to pass the BBB into the brain. AChE inhibitors that do not have charged moieties in their structure are considered potential drugs for the treatment of AD symptoms. Bisuracil 3, which can be expected to have the capability of crossing the BBB, may consequently be useful in the treatment of AD.

Herein, we describe the synthesis, in silico studies, and biological evaluation in vitro and in vivo of a series of new bisuracil scaffold-based AChE inhibitors—namely, the ionic and non-ionic counterparts **2** and **3**.

## 2. Results and Discussion

### 2.1. Synthesis of Bisuracils ***2*** and ***3***

The target bisuracils from a series **2** and **3** (Figure 2) with ionic and non-ionic amino-moieties were synthesised by starting from the same α,ω-{(3,6-dimethyluracil-1-yl)-[3-(5-bromopentyl)uracil-1-yl]}alkane (**4**) reagent. This bromide was first prepared from the reaction of 1-(ω-bromoalkyl)-3,6-dimethyluracil (**5a–d**) with 2,4-bis(trimethylsilyl)uracil (**6**) and alkylation of the resulting bis(uracil)alkane **7a–d** with 1,5-dibromopentane (Figure 1). The steps leading from bromide **4** to charged bisuracils **2a–d** included a replacement of the terminal Br atom in the bromide with a diethylamino-group and subsequent quaternisation of the *N* atom in amines **8a–d** with *o*-nitrile benzyl bromide (**9**). To produce non-ionic bisuracils **3a–d**, it is necessary to amine bromide **4** not with diethylamine as in the case of charged bisuracil **2a–d** preparation, but with ethylamine, resulting in the isolation of amines **10a–d**. Finally, the targeted non-ionic bisuracils **3a–d** were prepared via the alkylation of the secondary *N* atoms in the pentamethylene chains of amines **10a–d** with bromide **9** (Figure 2).

### 2.2. Inhibitory Effects on Cholinesterases of Counterparts ***2*** and ***3*** In Vitro

The inhibitory activities of bisuracils **2a–d** and **3a–d** against human AChE and BuChE for the 50% inhibitory concentration (IC_50_) values and their selectivity indexes for AChE over BuChE are presented in Table 1 and Table 2, respectively. In order to facilitate comparison of the data obtained for the new nitrile bisuracils **2a–d** with those for bisuracils with trifluoromethyl-l and nitro-groups, Table 1 provides a summary of the data published in an earlier report on the most active inhibitors from the series of ionic bisuracils **1a–f** [20]. The myasthenia drug pyridostigmine bromide for the charged AChE inhibitors in Table 1 and the AD drug donepezil for the non-ionic AChE inhibitors in Table 2 were used as reference drugs.

Ionic bisuracils **1a–f** have been shown to be highly effective and selective inhibitors of AChE, with the ability to inhibit the enzyme at picomolar or nanomolar concentrations. Additionally, these compounds have been observed to exhibit increased selectivity towards AChE compared to BuChE by more than 200,000 times. Bisuracils with CF_3_-substituents at benzene rings demonstrate greater selectivity towards AChE than bisuracils with NO_2_ substituents. In addition to the extreme selectivity towards AChE of bisuracils with CF_3_ and NO_2_ substituents, the introduction of nitrile substituents into the benzyl moiety further reduces IC_50_ values towards AChE and increases the selectivity for AChE vs. BChE. Thus, bisuracils **2a–d** with nitrile groups inhibit AChE at picomolar concentrations with selectivity for AChE over BuChE exceeding that of bisuracils with CF_3_ and NO_2_ groups by more than an order of magnitude. The reference drug inhibits AChE at a concentration hundreds of times higher than that of bisuracils and with a selectivity for AChE vs. BuChE of about three. The following can be observed in the change in inhibitory activity against AChE in the series of bisuracils **2a–d**. When the number of methylene groups in the spacer between 3,6-dimethyluracil moieties increased from three to four, the inhibitory activity against AChE and selectivity of AChE vs. BuChE increased. However, a lengthening of the spacer from four to five methylene groups, and further from five to six methylene groups, led to a decrease in inhibitory activity and selectivity. The bisuracils **2a,b** with three and four methylene groups in the spacer between 3,6-dimethyl and uracil moieties are unprecedentedly selective inhibitors of AChE vs. BuChE, with a selectivity of more than one million for bisuracil **2a** and more than three million for bisuracil **2b** (Table 1). To our knowledge, no AChE inhibitors with such extraordinary selectivity for the enzyme have previously been discovered.

Non-ionic bisuracils **3a–d** exhibit an inhibitory effect against AChE, with an IC_50_ more than an order of magnitude higher than that of ionic counterparts **2a–d** (Table 2). Meanwhile, their selectivity for AChE vs. BuChE is one to two orders of magnitude lower than the selectivity of bisuracils **2a–d**. Like bisuracils **2a–d**, as well as inhibiting AChE at nanomolar concentrations, the reference drug donepezil demonstrates more than an order of magnitude lower selectivity for AChE. The moderate 50-fold selectivity of galantamine for AChE over BuChE [24] and the non-selective inhibition profile of rivastigmine [25] sharply contrasts with the exceptional more than 10,000-fold AChE selectivity exhibited by bisuracils **3c,d**. This pronounced selectivity may significantly reduce off-target effects compared to conventional cholinesterase inhibitors.

In the series of non-ionic AChE inhibitors **3a–d**, there is no clear relationship between the number of methylene groups in the bridge between the uracil moieties and the IC_50_ or the selectivity with respect to AChE. For bisuracils **3a–d**, with a number of methylene groups in the bridge from three to six, IC_50_ changes in the range of 2.8–7.9 nM, while the selectivity against AChE vs. BChE changes from 6090 to 17,639 (Table 2).

Based on the data from the study of inhibitory activity in a series of ionic and non-ionic AChE inhibitors, bisuracils **2b** and **3c** were selected as lead compounds. Exhibiting the lowest values of IC_50_ and the highest selectivity for AChE in comparison with BuChE, this series of compounds also offers better prospects for AChE inhibition in vivo.

In order to study the mechanism of AChE and BuChE inhibition by bisuracils **2b** and **3c**, the inhibition constants (*K*_i_) were determined. Experiments were performed using four different concentrations of ATCh as a substrate for AChE, and with BuTCh as a substrate for BuChE. According to the analysis of Dixon and Cornish–Bowden plots for bisuracil **2b**, AChE and BuChE inhibition is of a mixed type characterised both by competitive inhibition (*K*_ci_) and uncompetitive inhibition (*K*_ui_) components. For AChE, *K*_ci_ was determined to be 55.7 ± 8.1 pM while *K*_ui_ = 156 ± 18 pM (Figure 3A,B), and *K*_ci_ = 0.13 ± 0.02 mM while *K*_ui_ = 0.7 ± 0.06 mM for BuChE (Figure 3C,D). Inhibition of AChE and BuChE by bisuracil **3c** was of a mixed type. For the AChE constant, *K*_ci_ was 0.7 ± 0.04 nM while *K*_ui_ 4.45 ± 1.76 nM (Figure 4A,B); for the BuChE constant, *K*_ci_ = 0.04 ± 0.007 мM while *K*_ui_ = 0.68 ± 0.08 мM (Figure 4C,D).

### 2.3. In Vivo Biological Assays

#### 2.3.1. Acute Toxicity

The median lethal doses (value of LD_50_) for mice of the lead compounds were calculated to be 5.16 mg/kg (95% confidence limits: 3.68–6.94 mg/kg) for bisuracil **2b** and 22.8 mg/kg (95% confidence limits: 15.04–28.73 mg/kg) for bisuracil **3c**.

#### 2.3.2. Pre-Treatment of Poisoning with OPs

The concept of using reversible AChE inhibition as a preventive strategy to enhance the efficacy of traditional antidotes in OP poisoning is a well-established principle [5,26]. This study explores the potential of bisuracil **2b** as a preventive agent to protect AChE from irreversible inhibition caused by paraoxon (POX), a model OP. Mice were pre-treated with bisuracil **2b** via intraperitoneal (i.p.) injection at different doses 30 min before being exposed to 2xLD_50_ POX (0.45 mg/kg). Following POX administration, the standard antidote atropine was given at a dose of 15 mg/kg within one minute. The protective effects of bisuracil **2b** are summarised in Table 3. The most effective protective effect of bisuracil **2b** was observed with a dose of 0.01 mg/kg at which 10 out of 12 mice survived. Increasing the dose of bisuracil **2b** reduced the protective effect in POX poisoned animals (Table 3).

We also estimated the relative risk of death (RRD) as a function of precocity (from min to hours) over a period of 10 h by Cox survival analysis [27]. Statistical differences in the number of surviving animals between experimental groups were tested using the Cox–Mantel multisample logarithmic rank criterion [27,28]. It was shown that, in mice exposed to 0.42 mg/kg of POX, RRD = 1 (Figure 5). However, in animals that received atropine within 1 min after poisoning, POX-induced mortality was lower (RRD = 0.5; Figure 5).

In mice pre-treated with bisuracil **2b** at a dose of 0.01 mg/kg, the RRD significantly decreased as compared to the poisoned group treated only with atropine (*p* = 0.02). However, further increases in the dose of bisuracil **2b** did not result in a reduction in the RRD (Figure 5). Therefore, it can be concluded that bisuracil **2b** significantly reduced mortality in mice at the dose of 0.01 mg/kg.

#### 2.3.3. Inhibition of Brain AChE

Bisuracil **3c** was identified as the most promising for the inhibition of brain AChE in vivo. To evaluate its inhibitory potential, we measured AChE activity in mouse brain homogenates at 30 min after i.p. injection of **3c** at doses of 0.5 LD_50_ (10 mg/kg) and 0.25 LD_50_ (5 mg/kg). Bisuracil **3c** was observed to inhibit AChE activity in the brain by 54.2 ± 6.6% at a dose of 10 mg/kg and 14.9 ± 3.8% at a dose of 5 mg/kg as compared to the mean AChE activity in the brains of control mice. Consequently, **3c** has been demonstrated to effectively inhibit brain AChE, rendering it a promising candidate for the treatment of memory impairment associated with AD.

#### 2.3.4. Behavioural Test

This study explored the ability of bisuracil **3c** to counteract scopolamine-induced amnesia in mice, utilising behavioural assessments conducted through the novel object recognition test. The test operates on the principle that animals demonstrating a significant preference for exploring new objects, as opposed to familiar ones, can be considered evidence of intact recognition memory [29]. The induction of amnesia was facilitated by the i.p. injection of scopolamine (1.5 mg/kg) 50 min prior to testing. Subsequently, donepezil (1 mg/kg, i.p.) or bisuracil **3c** (5 or 2 mg/kg, i.p.) was administered 30 min prior to testing. An equivalent amount of the vehicle was administered to a control group of mice.

It was shown that the mice in the control group preferred a novel object over a familiar object with a probability of 66.0 ± 2.9%. Administration of scopolamine at a dose of 1.5 mg/kg significantly reduced the preference for the novel object by 34.1% as compared to the control group (*p* = 0.0001). In contrast, the administration of bisuracil **3c** at a dose of 5 mg/kg served to prevent scopolamine-induced memory impairment. It was demonstrated that the administration of bisuracil 3c resulted in a significant 30.5% increase in the preference for the novel object compared to the scopolamine-treated group (*p* = 0.01) (Figure 6). Concurrently, the values did not deviate from those of the control group of animals (*p* = 0.975). The administration of donepezil at a dose of 1 mg/kg similarly resulted in a significant 32.2% (*p* = 0.03) increase in the preference for the novel object over the familiar one as compared to the group receiving scopolamine. Notably, bisuracil **3c** at a dose of 2 mg/kg had no significant effect on the preference index (Figure 6).

Thus, memory deficits were improved when either compound bisuracil **3c** at a dose of 5 mg/kg or donepezil at a dose of 1 mg/kg was administered.

#### 2.3.5. Effect on the Level of Locomotor Activity in Mice

The ability of effective doses of bisuracil **2b** (0.01 mg/kg) and bisuracil **3c** (5 mg/kg) to induce incapacitation was evaluated. Locomotor activity, measured as the total distance travelled, and exploratory behaviour, assessed by the number of rearings (head inclinations) and head dips, were evaluated using the open-field test. Additionally, motor coordination was analysed using the rotarod test, where the ability of mice to stay on the rotating rod was recorded. The compounds were administered i.p. 30 min before testing, while the control group received an equivalent volume of the vehicle.

The results showed that both groups of **2b**- and **3c**-treated mice explored the open field without any reduction in locomotor activity. As a result, the distance travelled by mice in both experimental groups was comparable to that of the control group, showing no significant differences (Figure 7). Furthermore, this study revealed that neither of the tested compounds significantly influenced the number of rearings or head dips (Figure 7B,C). The rotarod test results indicated that mice treated with bisuracils **2b** and **3c** exhibited no signs of motor impairment. The mean fall time of mice receiving **2b** (0.01 mg/kg, i.p.) was 124 ± 20 s, which was not significantly different from the control group, whose mean fall time was 173 ± 21 s (*p* = 0.125) (Figure 7D). A similar outcome was observed in mice administered **3c** at the maximum dose (5 mg/kg, i.p.), whose mean fall time was 170 ± 19 s, which was not significantly different from the control group, whose mean fall time was 154 ± 20 s (*p* = 0.137) (Figure 7D).

Consequently, the open-field and rotarod tests showed no impairments in locomotor or exploratory activity in mice following exposure to both bisuracils **2b** and **3c**. The results indicated that the tested compounds did not induce negative effects on movement coordination and stereotyped exploratory behaviour in mice.

### 2.4. Computational Modelling

#### 2.4.1. Molecular Docking and Molecular Dynamics

Molecular docking and molecular dynamics (MD) simulations were performed to compare docking scores and ligand–receptor interactions [30,31]. This analysis aims to provide deeper insights into the structure–activity relationships of bisuracils, thereby guiding subsequent structural optimisation efforts and facilitating the identification of superior lead compounds. For this reason, we used MOE for molecular docking simulation of the lead compounds **2b** and **3c**. The docking scoring results are shown in Table 4.

To evaluate the binding mode and interaction between bisuracils **2b**, **3c**, and AChE, the title compounds were docked into the crystal structure of *h*AChE complexed with donepezil (PDB: 7E3H). The detailed interactions between ligands and receptors are shown in Table 5. Figure 8A shows a possible binding mode and binding site for bisuracil **2b** (IC_50_ = 64.9 pM), while Figure 8B indicates the interaction between bisuracil **2b** and receptor residues. Bisuracil **2b** forms close interactions with residues Ser293, Val294, and Phe295 through three hydrogen bonds; C5′H acts as a hydrogen bond donor forming a hydrogen bond with Ser293 (distance = 3.40 Å), while carbonyl oxygen in C4′=O of pyrimidine ring acts as a hydrogen bond acceptor interacting with Val294 and Phe295 (distances = 3.30 Å, 3.08 Å), respectively. Bisuracil **2b** also has extensive H-pi interactions between C10H, C10′H, and the ethyl group at quaternised N with residues Trp286, Tyr341, and Tyr337, while C10H interacts twice with Trp286 (distance = 3.94 Å, 3.69 Å, 3.93 Å, and 4.27 Å, respectively), and uracil and benzene rings interact with Tyr341 and His447 by pi-H interaction (distance = 3.92 Å, 3.46 Å). The 3,6-dimethyluracil cycle of **2b** extends into the solvent space outside the binding pocket to interact with the solvent. Moreover, bisuracil **2b**, which occupies PAS, has strong interactions with key residues Trp286, Tyr341, and Phe295. Furthermore, **2b** interacts with His447 of the CAS catalytic triad, as well as residues of the anion-binding hydrophobic sites Tyr341 and Tyr337. These unique interactions play a crucial role in its excellent activity.

In comparison with charged bisuracil **2b**, Figure 8C demonstrates a possible binding mode and binding site for non-ionic bisuracil **3c** (IC_50_ = 2.78 nM), while Figure 8D indicates the interaction between **3c** and receptor amino acid residues. Bisuracil **3c** forms interactions with residues Asp74 and Tyr133 through hydrogen bonds. C7′H acts as a hydrogen bond donor forming a hydrogen bond with Asp74 (distance = 3.38 Å), while C≡N acts as a hydrogen bond acceptor interacting with Tyr133 (distances = 2.79Å). In addition, bisuracil **3c** also has extensive H-pi interactions of C5′H, C9′H, and C7H with residues Tyr337, Tyr341, and Trp286 (distance = 4.29 Å, 4.78 Å, and 3.95 Å, respectively); the benzene ring interacts twice with Trp86 by pi-H interaction (distance = 3.73 Å, 4.12 Å) and with residues Tyr341 by pi-pi interaction (distance = 3.48 Å). The 3,6-dimethyluracil cycle of **3c** extends into the solvent space outside the binding pocket and interacts with the solvent.

According to the results of molecular docking, the flexible polymethylene chain allows the title bisuracils to bind both key sites CAS and PAS of AChE. The aromatic ring mainly interacts with the CAS site, while the uracil ring and pentamethylene chain favour interactions with the key residues of the PAS site. The 3,6-dimethyluracil cycle extends outward to interact with solvents. This unique binding mode enables this kind of bisuracil to exhibit excellent AChE inhibitory activity.

Mixed-type inhibition suggests simultaneous binding at CAS and PAS, corroborated by docking (Figure 8A–D). This dual binding may sterically hinder substrate access, enhancing inhibition.

In order to explore the details of the **2b** and **3c** title compounds binding to cholinesterases and further structural modifications, two series of representative bisuracils **2b** and **3c** were subjected to 100 ns MD simulations with AChE and BuChE, respectively. The root-mean-square deviations (RMSDs) of the bisuracils **2b** and **3c** were calculated to evaluate the stability and convergence of the titled bisuracil AChE and BuChE complex during the MD simulations (Figure 9 and Figure 10).

Figure 10 illustrates the critical role played by the quaternary ammonium moiety and terminal 3,6-dimethyluracil cycle in the binding process; the carbonyl group on the central uracil ring also significantly contributes to the interaction. The RMSD analysis from MD simulations provided insights into the stability of bisuracils **2b** and **3c** binding to AChE and BuChE. For both compounds, the trajectory fluctuations were notably smaller when binding to AChE than those observed for BuChE (Figure 9 and Figure 10). This characteristic allowed them to more rapidly reach a steady state, as well as maintaining a more stable binding mode when interacting with AChE. Notably, bisuracil **2b** exhibits superior stability compared to compound **3c** when bound to AChE. The smoother trajectory of **2b** assumes a more consistent and durable interaction within the enzyme’s active site. This enhanced stability was corroborated by its excellent performance in molecular docking studies, which further rationalised its superior activity relative to bisuracil **3c**. The significant difference in the binding stability between AChE and BuChE for **2b** and **3c** underscored the structural distinctions between these enzymes. Bisuracil **2b**’s pronounced selectivity for AChE over BuChE can be attributed to its optimal fit within the active site of AChE, resulting in higher affinity and lower IC_50_ values.

#### 2.4.2. ADMET Prediction

ADMET refers to the Absorption, Distribution, Metabolism, Excretion, and Toxicity of compounds in the human body. The ADMET prediction results (Table 6) indicated that synthesised bisuracils **2a–d** and **3a–d** have relatively good safety profiles, with a low probability of acute oral toxicity, and possess good lipophilicity and transmembrane ability, which enables them to penetrate the blood–brain barrier effectively and exert their effects in the brain.

## 3. Materials and Methods

### 3.1. Chemistry

#### 3.1.1. General Information

The NMR experiments were carried out on Bruker spectrometers AVANCE-400 (400.1 MHz (^1^H), 100.6 MHz (^13^C)). Chemical shifts (δ in ppm) were referenced to the solvents DMSO-d_6_ (δ = 2.50 ppm for ^1^H and 40.0 ppm for ^13^C NMR) and CDCl_3_ (δ = 7.26 ppm for ^1^H and 77.0 ppm for ^13^C NMR). MALDI-TOF mass spectra were recorded on a Bruker ULTRAFLEX III mass spectrometer using *p*-nitroaniline or 2,5-dihydroxybenzoic acid as a matrix, and conditions of mass spectrum recording: Nd:YAG laser, λ = 355 nm, repetition rate 100 Hz, and linear mode with the registration of positively or negatively charged ions. Microanalyses of C, H, and N were performed with a CHNS analyser Vario Macro cube (Elementar Analysensysteme GmbH, Germany); they were within ±0.3% of theoretical values for C, H, and N. Thin-layer chromatography was performed on Silufol-254 plates (the solvent system is diethyl ether or an ethylacetate–methanol mixture); visualisation of spots was carried out under UV light (λ = 254 nm). For column chromatography, silica gel of 60 mesh from Fluka was used. All solvents were dried according to standard protocols.

Procedures for the preparation of α,ω-{(3,6-dimethyluracil-1-yl)-[3-(5-bromopentyl)uracil-1-yl]}alkanes (**4a–d**) and α,ω-{(3,6-dimethyluracil-1-yl)-[3-(5-diethylaminopentyl)uracil-1-yl]}alkanes (**8a–d**) from **4a–d** were as described previously [20].

#### 3.1.2. Synthesis of Charged Bisuracils with Nitrile Substituent

##### General Procedure to Obtain Bisuracils **2a–d**

A solution of amine **8a–d** (1.0 mmol) and 1.1-fold excess *o*-nitrilebenzyl bromide **9** in MeCN (30 mL) was refluxed for 16 h at 60–65 °C. The solvent was distilled off and the residue was triturated in anhydrous diethyl ether (2 × 30 mL), each time decanted after settling of the compound. Water (70 mL) and activated carbon (0.1–0.15 g, powder) were added to the residue. The suspension was stirred for 1 h at 35–40 °C and filtered after cooling to room temperature. The solvent was evaporated, and the residue was dissolved in CHCl3 (200 mL) and dried with MgSO_4_ to afford the titled bisuracil as a hygroscopic light brown powder with an unclear melting point in the temperature range of 40–60 °C.

**1,3-{(3,6-Dimethyluracil-1-yl)-[3-(5-*o*-nitrilebenzyl(diethyl)ammonio)pentyl)-uracil-1-yl]}propane bromide (2a)**: 0.42 g, 67%; ^1^H NMR (CDCl_3_, 400 MHz) δ 8.32 (d, *J* = *7.6* Hz, 1H, C3′′H), 7.83–7.79 (m, 2H, C4′′H, C5′′H), 7.68–7.65 (m, 1H, C6′′H), 7.38 (d, *J* = *7.6* Hz,1H, C6′H), 5.72 (d, *J* = *7.6* Hz, 1H, C5′H), 5.63 (s, 1H, C5H), 5.17 (br. s, 2H, NCH_2_Ph), 3.94–3.86 (m, 6H, C7H_2_, C9H_2_, C7′CH_2_), 3.72–3.70 (m, 4H, 2NCH_2_), 3.55 (m, 2H, C11′H_2_), 3.29 (s, 3H, N3CH_3_), 2.25 (s, 3H, C6CH_3_), 2.09 (m, 2H, C8H_2_), 1.85 (m, 2H, C10′H_2_), 1.69 (m, 2H, C8′H_2_), 1.25–1.22 (m, 8H, C9′H_2_, 2CH_3_); ^13^C NMR (CDCl_3_, 100 MHz) δ 163.0 (C′4), 162.2 (C4), 152.7 (C6′), 152.4 (C6), 151.4 (C2), 150.8 (C2′), 142.7 (C2′′), 135.5 (C5′′), 134.3 (C1′′), 133.9 (C6′′), 131.5 (C4′′), 130.9 (C3′′), 101.9, 101.7 (C5′, C5), 61.2 (C11′), 59.2 (NCH_2_Ph), 55.2 (NCH_2_CH_3_), 47.5 (C9), 42.3 (C7), 40.0 (C7′), 28.5 (C10′), 27.9 (N3CH_3_), 26.6 (C8′), 23.4 (C8), 22.2 (C9′), 19.9 (C6CH_3_), 9.0 (NCH_2_CH_3_); MALDI-MS [M − H]^+^, [M − Br + H]^+^ calcd for C_30_H_41_BrN_6_O_4_
*m/z* 627.2, 549.3, respectively. Found: 627.2, 549.3. Anal. calcd for C_30_H_41_BrN_6_O_4_: C, 57.23; H, 6.56; Br, 12.69; N, 13.35. Found: C, 57.31; H, 6.46; Br, 12.81; N, 13.28 (Appendix A).

**1,4-{(3,6-Dimethyluracil-1-yl)-[3-(5-*o*-nitrilebenzyl(diethyl)ammonio)pentyl)-uracil-1-yl]}butane bromide (2b)**: 0.46 g, 71%; ^1^H NMR (CDCl_3_, 400 MHz) δ 8.17–8.14 (m, 1H, C3′′H), 7.83–7.80 (m, 2H, C4′′H, C5′′H), 7.70–7.67 (m, 1H, C6′′H), 7.39 (d, *J* = *6.0* Hz,1H, C6′H), 5.68 (d, *J* = *7.6* Hz, 1H, C5′H), 5.58 (s, 1H, C5H), 5.17 (br. s, 2H, NCH_2_Ph), 3.89–3.82 (m, 6H, C7H_2_, C10H_2_, C7′CH_2_), 3.65 (m, 4H, 2NCH_2_), 3.46 (m, 2H, C11′H_2_), 3.27 (s, 3H, N3CH_3_), 2.26 (s, 3H, C6CH_3_), 1.87 (m, 2H, C10′H_2_), 1.75–1.68 (m, 6H, C8H_2_, C9H_2_, C8′H_2_), 1.48–1.41 (m, 8H, C9′H_2_, 2CH_3_); ^13^C NMR (CDCl_3_, 100 MHz) δ 162.8 (C′4), 160.0 (C4), 152.0 (C6′), 151.1 (C6), 150.9 (C2), 150.8 (C2′), 142.8 (C2′′), 134.8 (C5′′), 134.0 (C1′′), 133.8 (C6′′), 131.3 (C4′′), 130.0 (C3′′), 101.3, 101.2 (C5′, C5), 60.5 (C11′), 58.7 (NCH_2_Ph), 54.8 (NCH_2_CH_3_), 46.6 (C10), 44.1 (C7), 39.7 (C7′), 27.5 (C10′), 25.8 (N3CH_3_), 26.6 (C8′), 23.6, 23.2 (C8, C9), 22.3 (C9′), 19.6 (C6CH_3_), 8.7, 8.5 (NCH_2_CH_3_); MALDI-MS [M − H]^+^, [M − Br + H]^+^ calcd for C_31_H_43_BrN_6_O_4_
*m/z* 641.3, 563.3, respectively. Found: 641.3, 563.4. Anal. calcd for C_31_H_43_BrN_6_O_4_: C, 57.85; H, 6.73; Br, 12.41; N, 13.06. Found: C, 57.71; H, 6.70; Br, 12.31; N, 13.18 (Appendix A).

**1,5-{(3,6-Dimethyluracil-1-yl)-[3-(5-*o*-nitrilebenzyl(diethyl)ammonio)pentyl)-uracil-1-yl]}pentane bromide (2c)**: 0.55 g, 84%; ^1^H NMR (CDCl_3_, 400 MHz) δ 8.26 (m, 1H, C3′′H), 7.78–7.75 (m, 2H, C4′′H, C5′′H), 7.65–7.62 (m, 1H, C6′′H), 7.20 (d, *J* = *6.4* Hz,1H, C6′H), 5.67 (d, *J* = *7.4* Hz, 1H, C5′H), 5.57 (s, 1H, C5H), 5.14 (br. s, 2H, NCH_2_Ph), 3.88 (m, 2H, C7′H_2_), 3.77–3.69 (m, 8H, C11H_2_, C7CH_2_, 2NCH_2_), 3.48 (m, 2H, C11′H_2_), 3.26 (s, 3H, N3CH_3_), 2.23 (s, 3H, C6CH_3_), 1.81 (m, 2H, C10′H_2_), 1.67 (m, 6H, C8H_2_, C10H_2_, C8′H_2_), 1.48–1.41 (m, 10H, C9H_2_, C9′H_2_, 2CH_3_); ^13^C NMR (CDCl_3_, 100 MHz) δ 163.1 (C′4), 162.3 (C4), 152.2 (C6′), 151.3 (C6), 151.1 (C2, C2′), 142.6 (C2′′), 135.5 (C5′′), 134.2 (C1′′), 131.4 (C6′′), 130.9 (C4′′, C3′′), 101.4 (C5′, C5), 61.1 (C11′), 59.1 (NCH_2_Ph), 55.2 (NCH_2_CH_3_), 49.4 (C11), 44.7 (C7), 39.9 (C7′), 28.4 (C10′), 28.2, 27.7 (C8, C8′), 26.6 (C10, N3CH_3_), 23.5 (C9′), 22.2 (C9), 19.8 (C6CH_3_), 9.0 (NCH_2_CH_3_); MALDI-MS [M − Br]^+^ calcd for C_32_H_45_BrN_6_O_4_
*m/z* 577.4, respectively. Found: 577.4. Anal. calcd for C_32_H_45_BrN_6_O_4_: C, 58.44; H, 6.90; Br, 12.15; N, 12.78. Found: C, 58.35; H, 7.00; Br, 12.06; N, 12.83 (Appendix A).

**1,6-{(3,6-Dimethyluracil-1-yl)-[3-(5-*o*-nitrilebenzyl(diethyl)ammonio)pentyl)-uracil-1-yl]}hexane bromide (2d)**: 0.60 g, 90%; ^1^H NMR (CDCl_3_, 400 MHz) δ 8.37–8.35 (d, *J* = *7.6* Hz, 1H, C3′′H), 7.83–7.78 (m, 2H, C4′′H, C5′′H), 7.66–7.63 (m, 1H, C6′′H), 7.16–7.14 (d, *J* = *8.4* Hz,1H, C6′H), 5.72–5.69 (d, *J* = *7.6* Hz, 1H, C5′H), 5.62 (s, 1H, C5H), 5.18 (br. s, 2H, NCH_2_Ph), 3.92 (m, 2H, C7′H_2_), 3.82–3.78 (m, 2H, C12H_2_), 3.73 (m, 6H, C7CH_2_, 2NCH_2_), 3.54 (m, 2H, C11′H_2_), 3.31 (s, 3H, N3CH_3_), 2.24 (s, 3H, C6CH_3_), 1.82 (m, 2H, C10′H_2_), 1.70 (m, 6H, C8H_2_, C11H_2_, C8′H_2_), 1.48–1.41 (m, 12H, C9H_2_, C10H_2_, C9′H_2_, 2CH_3_); ^13^C NMR (CDCl_3_, 100 MHz) δ 162.9 (C′4), 162.4 (C4), 152.3 (C6′), 151.3 (C6), 150.9 (C2, C2′′), 142.1 (C2′), 137.7 (C5′′), 134.4 (C1′′), 132.2 (C6′′), 125.8 (C4′′), 122.1 (C3′′), 101.7, 101.6 (C5′, C5), 59.0 (C11′), 58.7 (NCH_2_Ph), 55.3 (NCH_2_CH_3_), 49.6 (C7), 46.6, 45.0 (C12, C7′), 28.9 (C10′), 27.8 (N3CH_3_), 27.1 (C8′), 26.4 (C9′), 26.3, 26.0 (C8, C11), 23.0, 22.9 (C9, C10), 19.8 (C6CH_3_), 9.0, 8.6 (NCH_2_CH_3_); MALDI-MS [M − H]^+^, [M − Br + H]^+^ calcd for C_33_H_47_BrN_6_O_4_
*m/z* 669.3, 591.4, respectively. Found: 669.3, 591.4. Anal. calcd for C_33_H_47_BrN_6_O_4_: C, 59.01; H, 7.05; Br, 11.90; N, 12.51. Found: C, 58.95; H, 7.12; Br, 12.02; N, 12.60 (Appendix A).

#### 3.1.3. Synthesis of Non-Ionic Bisuracils with Nitrile Substituent

##### General Procedure to Replace Atom of Br in Bromides **4a–d** with Ethylamino-Group

A 20% EtNH_2_ solution in *i*-PrOH (2.5 mL, ca. 11.0 mmol) was added to the mixture of bromide **4a–d** (2.0 mmol) and potassium carbonate (0.28 g, 2.0 mmol) in MeCN (50 mL), and the mixture was stirred at 35–40 °C for 8 h. A precipitate was filtered out, and the solvent was evaporated in vacuo to afford the targeted amine as a clear light yellow oil.

**1,3-{(3,6-Dimethyluracil-1-yl)-[3-(5-ethylaminopentyl)uracil-1-yl]}propane (10a)**: 0.77 g, 95%; ^1^H NMR (CDCl_3_, 400 MHz) δ 7.20 (d, *J* = 6.4 Hz, 1H, C6′H), 5.70 (d, *J* = 7.2 Hz, 1H, C5′H), 5.58 (s, 1H, C5H), 3.89–3.84 (m, 6H, C7′H_2_, C9H_2_), 3.80–3.77 (m, 2H, C7H_2_), 3.27 (s, 3H, N3CH_3_), 2.63–2.56 (m, 4H, C11′H_2_, NCH_2_), 2.24 (s, 1H, NH), 2.21 (s, 3H, C6CH_3_), 2.07–2.01 (m, 2H, C8H_2_), 1.62–1.56 (m, 2H, C8′H_2_); 1.54–1.48 (m, 2H, C10′H_2_), 1.37–1.30 (m, 2H, C9′H_2_), 1.09–1.06 (t, *J* = *6.0* Hz, 3H, CH_3_); ^13^C NMR (CDCl_3_, 100 MHz) δ 162.7 (C′4), 162.0 (C4), 154.0 (C6′), 152.3 (C6), 150.4 (C2), 141.9 (C2′), 102.0 (C5′, C5), 49.3 (C11′), 47.3 (NCH_2_CH_3_), 43.9, 42.1 (C7, C9), 41.1 (C7′), 29.4 (C10′), 28.5 (C8′), 27.8 (N3CH_3_), 27.3 (C8), 24.6 (C9′), 19.5 (C6CH_3_), 14.9 (NCH_2_CH_3_); ESI-MS [M + H]^+^ calcd for C_20_H_31_N_5_O_4_
*m/z* 406.3. Found: 406.3. Anal. calcd for C_20_H_31_N_5_O_4_: C, 59.24; H, 7.71; N, 17.27. Found: C, 59.14; H, 7.62; N, 17.21 ((Appendix A).

**1,4-{(3,6-Dimethyluracil-1-yl)-[3-(5-ethylaminopentyl)uracil-1-yl]}butane (10b)**: 0.81 g, 96%; ^1^H NMR (CDCl_3_, 400 MHz) δ 7.13 (d, *J* = 6.4 Hz, 1H, C6′H), 5.67 (d, *J* = 6.0 Hz, 1H, C5′H), 5.57 (s, 1H, C5H), 3.88–3.85 (t, *J* = 6.0 Hz, 2H, C7′H_2_), 3.84–3.81 (t, *J* = 6.0 Hz, 1H, C10H_2_), 3.76–3.73 (t, *J* = 5.6 Hz, 2H, C7H_2_), 3.26 (s, 3H, N3CH_3_), 2.71–2.59 (m, 5H, C11′H_2_, NCH_2_, NH), 2.21 (s, 1H, NH), 2.21 (s, 3H, C6CH_3_), 1.75–1.52 (m, 8H, C8H_2_, C8′H_2_, C9H_2_, C10′H_2_), 1.36–1.30 (m, 2H, C9′H_2_), 1.11–1.08 (t, *J* = *5.6* Hz, 3H, CH_3_); ^13^C NMR (CDCl_3_, 100 MHz) δ 162.9 (C′4), 162.2 (C4), 152.3 (C6′), 151.3 (C6), 150.7 (C2), 142.7 (C2′), 101.8 (C5′, C5), 49.0 (C11′), 44.2 (NCH_2_CH_3_), 43.7, 41.0 (C7, C10, C7′), 28.9 (C10′), 27.7 (C8′), 27.2 (N3CH_3_), 26.0, 25.7 (C8, C9), 24.5 (C9′), 19.6 (C6CH_3_), 14.4 (NCH_2_CH_3_); MALDI-MS [M + H]^+^ calcd for C_21_H_33_N_5_O_4_
*m/z* 420.3. Found: 420.1. Anal. calcd for C_21_H_33_N_5_O_4_: C, 60.12; H, 7.93; N, 16.69. Found: C, 60.18; H, 8.00; N, 16.58 (Appendix A).

**1,5-{(3,6-Dimethyluracil-1-yl)-[3-(5-ethylaminopentyl)uracil-1-yl]}pentane (10c)**: 0.80 g, 92%; ^1^H NMR (CDCl_3_, 400 MHz) δ 7.06 (d, *J* = 6.0 Hz, 1H, C6′H), 5.65 (d, *J* = 6.4 Hz, 1H, C5′H), 5.53 (s, 1H, C5H), 3.87–3.84 (t, *J* = 6.0 Hz, 2H, C7′H_2_), 3.76–3.73 (t, *J* = 6.4 Hz, 1H, C11H_2_), 3.70–3.67 (t, *J* = 6.0 Hz, 2H, C7H_2_), 3.24 (s, 3H, N3CH_3_), 2.61–2.53 (m, 4H, C11′H_2_, NCH_2_), 2.21 (s, 3H, C6CH_3_), 1.92 (s, 1H, NH), 1.70–1.63 (m, 4H, C8H_2_, C10H_2_), 1.59–1.55, 1.49–1,46 (both m, 2H each, C8′H_2_, C10′H_2_), 1.37–1.30 (m, 4H, C9H_2_, C9′H_2_), 1.06–1.03 (t, *J* = *5.6* Hz, 3H, CH_3_); ^13^C NMR (CDCl_3_, 100 MHz) δ 162.8 (C′4), 162.2 (C4), 152.1 (C6′), 151.2 (C6), 150.7 (C2), 141.8 (C2′), 101.6 (C5′, C5), 49.4 (C11′), 49.2 (NCH_2_CH_3_), 44.6, 43.9, 41.0 (C7, C11, C7′), 29.5 (C10′), 28.4 (C8′), 28.1 (N3CH_3_), 27.7, 27.3 (C8, C10), 24.6, 23.3 (C9, C9′), 19.5 (C6CH_3_), 15.0 (NCH_2_CH_3_); MALDI-MS [M + H]^+^ calcd for C_22_H_35_N_5_O_4_
*m/z* 434.3. Found: 434.4. Anal. calcd for C_22_H_35_N_5_O_4_: C, 60.95; H, 8.14; N, 16.15. Found: C, 61.04; H, 8.03; N, 16.23 (Appendix A).

**1,6-{(3,6-Dimethyluracil-1-yl)-[3-(5-ethylaminopentyl)uracil-1-yl]}hexane (10d)**: 0.86 g, 96%; ^1^H NMR (CDCl_3_, 400 MHz) δ 7.05 (d, *J* = 7.6 Hz, 1H, C6′H), 5.63 (d, *J* = 8.0 Hz, 1H, C5′H), 5.52 (s, 1H, C5H), 3.87–3.83 (t, *J* = 5.2 Hz, 2H, C7′H_2_), 3.75–3.72 (t, *J* = 6.0 Hz, 1H, C12H_2_), 3.67–3.64 (t, *J* = 6.4 Hz, 2H, C7H_2_), 3.24 (s, 3H, N3CH_3_), 2.60–2.53 (m, 4H, C11′H_2_, NCH_2_), 2.43 (s, 1H, NH), 2.18 (s, 3H, C6CH_3_), 1.66–1.46 (m, 8H, C8H_2_, C11H_2_, C8′H_2_, C10′H_2_), 1.36–1.30 (m, 6H, C9H_2_, C10H_2_, C9′H_2_), 1.07–1.04 (t, *J* = *7.2* Hz, 3H, CH_3_); ^13^C NMR (CDCl_3_, 100 MHz) δ 162.9 (C′4), 162.2 (C4), 152.1 (C6′), 151.2 (C6), 150.8 (C2), 142.0 (C2′), 101.4 (C5′, C5), 49.4 (C11′), 48.7 (NCH_2_CH_3_), 44.8, 43.4, 40.9 (C7, C12, C7′), 28.7 (C10′), 28.5 (C8′), 28.4 (N3CH_3_), 27.6, 27.1 (C8, C11), 26.1, 25.8, 24.4 (C9, C10, C9′), 19.5 (C6CH_3_), 14.0 (NCH_2_CH_3_); MALDI-MS [M + H]^+^ calcd for C_23_H_37_N_5_O_4_
*m/z* 448.3. Found: 448.3. Anal. calcd for C_23_H_37_N_5_O_4_: C, 61.72; H, 8.33; N, 15.65. Found: C, 61.64; H, 8.43; N, 15.73 (Appendix A).

##### Synthesis of Non-Ionic Cholinesterase Inhibitors Based on Bisuracils **3a–d**: General Procedure

A mixture of amine **10a**–**d** (1.0 mmol), *ortho*-nitrilebenzyl bromide **9** (0.2 g, 1.0 mmol), and potassium carbonate (0.14 g, 1.0 mmol) was stirred in MeCN (30 mL) at 45–50 °C for 8 h. The precipitate was filtered out. The solution was concentrated to 10–15 mL and transferred to a column with SiO_2_. The column was successively washed with petroleum ether, ethyl acetate, and 10:1 ethyl acetate diethylamine mixture. The target bisuracils **3a**–**d** were isolated from the 10:1 ethyl acetate diethylamine mixture fractions as clear light brown oil.

**1,3-{(3,6-Dimethyluracil-1-yl)-[3-(5-*o*-nitrilebenzyl(ethyl)amino)pentyl)-uracil-1-yl]}propane (3a)**: 0.42 g, 80%; ^1^H NMR (CDCl_3_, 400 MHz) δ 7.56–7.51 (m, 2H, C3′′H, C4′′H), 7.48–7.45 (m, 1H, C5′′H), 7.26–7.23 (m, 1H, C6′′H), 7.21 (d, *J* = *8.0* Hz,1H, C6′H), 5.64 (d, *J* = *8.0* Hz, 1H, C5′H), 5.53 (s, 1H, C5H), 3.83–3.74 (m, 6H, C7H_2_, C9H_2_, C7′CH_2_), 3.66 (br. s, 2H, NCH_2_Ph), 3.21 (s, 3H, N3CH_3_), 2.49–2.44, 2.40–2.37 (both m, 2H each, NCH_2_, C11′H_2_), 2.16 (s, 3H, C6CH_3_), 2.01–1.98 (m, 2H, C8H_2_), 1.54–1.48 (m, 2H, C10′H_2_), 1.46–1.40 (m, 2H, C8′H_2_), 1.26–1.20 (m, 2H, C9′H_2_), 0.97–0.94 (t, *J* = *6.8* Hz, 3H, CH_3_); ^13^C NMR (CDCl_3_, 100 MHz) δ 162.6 (C′4), 162.0 (C4), 152.1 (C6′), 151.2 (C6), 150.5 (C2, C2′), 141.9 (C2′′), 132.5, 132.4 (C1′′, C5′′), 129.6 (C3′′, C4′′, C6′′), 101.7, 101.6 (C5′, C5), 55.9 (C11′), 52.9 (NCH_2_Ph), 47.2 (NCH_2_CH_3_), 42.0, 40.9 (C7, C7′, C9), 28.3 (C10′), 27.6 (N3CH_3_), 27.1 (C8′), 26.3 (C8), 24.4 (C9′), 19.4 (C6CH_3_), 11.3 (NCH_2_CH_3_); MALDI-MS [M − Et − CN + 2H]^+^, [M − H]^+^ calcd for C_28_H_36_N_6_O_4_
*m/z* 468.3, 520,3, respectively. Found: 468.2, 520.2. Anal. calcd for C_28_H_36_N_6_O_4_: C, 64.60; H, 6.97; N, 16.14. Found: C, 64.71; H, 7.06; Br, 12.81; N, 16.08 (Appendix A).

**1,4-{(3,6-Dimethyluracil-1-yl)-[3-(5-*o*-nitrilebenzyl(ethyl)amino)pentyl)-uracil-1-yl]}butane (3b)**: 0.43 g, 81%; ^1^H NMR (CDCl_3_, 400 MHz) δ 7.57–7.51 (m, 2H, C3′′H, C4′′H), 7.49–7.46 (m, 1H, C5′′H), 7.24–7.22 (m, 1H, C6′′H), 7.12 (d, *J* = *6.4* Hz,1H, C6′H), 5.62 (d, *J* = *6.0* Hz, 1H, C5′H), 5.51 (s, 1H, C5H), 3.81–3.77 (m, 4H, C10H_2_, C7′CH_2_), 3.73–3.70 (m, 2H, C7H_2_), 3.68 (br. s, 2H, NCH_2_Ph), 3.21 (s, 3H, N3CH_3_), 2.49–2.46, 2.41–2.38 (both m, 2H each, NCH_2_, C11′H_2_), 2.17 (s, 3H, C6CH_3_), 1.70–1.64 (both m, 2H each, C8H_2_, C9H_2_), 1.54–1.49, 1.45–1.41 (both m, 2H each, C8′H_2_, C10′H_2_), 1.26–1.20 (m, 2H, C9′H_2_), 0.98–0.95 (t, *J* = *6.0* Hz, 3H, CH_3_); ^13^C NMR (CDCl_3_, 100 MHz) δ 162.6 (C′4), 162.0 (C4), 152.1 (C6′), 151.2 (C6), 150.6 (C2, C2′), 142.0 (C2′′), 132.5, 132.4 (C1′′, C5′′), 129.7, 127.1 (C3′′, C4′′, C6′′), 101.5, 101.4 (C5′, C5), 55.9 (C11′), 52.9 (NCH_2_Ph), 48.7 (NCH_2_CH_3_), 47.2, 44.1, 40.8 (C7, C7′, C10), 27.6 (C10′), 27.1 (N3CH_3_), 26.2, 25.9, 25.6, 24.4 (C8, C9, C8′, C9′), 19.5 (C6CH_3_), 11.2 (NCH_2_CH_3_); MALDI-MS [M − H]^+^, [M − CN − 2H]^+^, [M − CH_2_PhCN + 2H]^+^, calcd for C_29_H_38_N_6_O_4_
*m/z* 533.3, 507.3, 420.3, respectively. Found: 533.4, 507.3, 420.2. Anal. calcd for C_29_H_38_N_6_O_4_: C, 65.15; H, 7.16; N, 15.72. Found: C, 65.11; H, 7.09; N, 15.58 (Appendix A).

**1,5-{(3,6-Dimethyluracil-1-yl)-[3-(5-*o*-nitrilebenzyl(ethyl)amino)pentyl)-uracil-1-yl]}pentane (3c)**: 0.43 g, 78%; ^1^H NMR (CDCl_3_, 400 MHz) δ 7.62–7.58 (m, 2H, C3′′H, C4′′H), 7.55–7.51 (m, 1H, C5′′H), 7.32–7.28 (m, 1H, C6′′H), 7.08 (d, *J* = *7.6* Hz,1H, C6′H), 5.70 (d, *J* = *7.6* Hz, 1H, C5′H), 5.58 (s, 1H, C5H), 3.91–3.87, 3.82–3.78 (both m, 2H each, C11H_2_, C7′CH_2_), 3.75–3.71 (m, 4H, C7H_2_, NCH_2_Ph), 3.30 (s, 3H, N3CH_3_), 2.56–2.51 (q, *J* = *7.2* Hz, 2H, NCH_2_), 2.47–2.43 (q, *J* = *7.2* Hz, 2H, C11′H_2_), 2.23 (s, 3H, C6CH_3_), 1.78–1.67 (both m, 2H each, C8H_2_, C10H_2_), 1.60–1.55, 1.51–1.46, 1.43–1.37, 1.35–1.29 (all m, 2H each, C8′H_2_, C10′H_2_, C9H_2_, C9′H_2_), 1.04–1.00 (t, *J* = *7.2* Hz, 3H, CH_3_); ^13^C NMR (CDCl_3_, 100 MHz) δ 162.8 (C′4), 162.2 (C4), 152.2 (C6′), 151.3 (C6), 150.7 (C2, C2′), 141.9 (C2′′), 132.6, 129.8, 127.2 (C1′′, C5′′, C3′′, C4′′, C6′′), 101.5 (C5′, C5), 56.0 (C11′), 53.0 (NCH_2_Ph), 49.2 (NCH_2_CH_3_), 47.4, 44.7, 41.0 (C7, C7′, C11), 28.4 (C10′), 28.2 (N3CH_3_), 27.7, 27.3, 24.6, 23.4 (C8, C9, C10, C8′, C9′), 19.5 (C6CH_3_), 11.2 (NCH_2_CH_3_); MALDI-MS [M − H]^+^ calcd for C_30_H_46_N_6_O_4_
*m/z* 547.3, respectively. Found: 547.3. Anal. calcd for C_30_H_46_N_6_O_4_: C, 65.67; H, 7.35; N, 15.32. Found: C, 65.58; H, 7.39; N, 15.18 (Appendix A).

**1,6-{(3,6-Dimethyluracil-1-yl)-[3-(5-*o*-nitrilebenzyl(ethyl)amino)pentyl)-uracil-1-yl]}hexane (3d)**: 0.45 g, 80%; ^1^H NMR (CDCl_3_, 400 MHz) δ 7.59–7.53 (m, 3H, C3′′H, C4′′H, C5′′H), 7.30 (m, 1H, C6′′H), 7.07 (d, *J* = *6.0* Hz,1H, C6′H), 5.67 (d, *J* = *6.0* Hz, 1H, C5′H), 5.56 (s, 1H, C5H), 3.89–3.86 (m, 2H, C7′CH_2_), 3.78–3.67 (m, 6H, C12H_2_, C7H_2_, NCH_2_Ph), 3.28 (s, 3H, N3CH_3_), 2.52–2.44 (m, 4H, NCH_2_, C11′H_2_), 2.21 (s, 3H, C6CH_3_), 1.63–1.50 (m, 8H, C8H_2_, C11H_2_, C8′H_2_, C10′H_2_), 1.37–1.26 (m, 6H, C9H_2_, C10H_2_, C9′H_2_), 1.02 (br. m, 3H, CH_3_); ^13^C NMR (CDCl_3_, 100 MHz) δ 162.9 (C′4), 162.3 (C4), 152.2 (C6′), 151.3 (C6), 150.7 (C2, C2′), 141.9 (C2′′), 132.6, 129.7, 127.1 (C1′′, C5′′, C3′′, C4′′, C6′′), 101.6 (C5′, C5), 56.0 (C11′), 53.1 (NCH_2_Ph), 49.5 (NCH_2_CH_3_), 47.4, 44.9, 41.0 (C7, C7′, C12), 28.8 (C10′), 28.6 (N3CH_3_), 27.7, 27.3, 26.2, 25.9, 24.6 (C8, C9, C10, C11, C8′, C9′), 19.6 (C6CH_3_), 11.5 (NCH_2_CH_3_); MALDI-MS [M + Na]^+^, [M − H]^+^, [M − CH_2_PhCN + 2H]^+^, calcd for C_31_H_42_N_6_O_4_
*m/z* 585.3, 561.3, 448.3, respectively. Found: 585.3, 561.4, 448.2. Anal. calcd for C_31_H_42_N_6_O_4_: C, 66.17; H, 7.52; N, 14.94. Found: C, 66.00; H, 7.59; N, 15.03 (Appendix A).

### 3.2. Biological Studies

#### 3.2.1. General Information

Paraoxon, atropine, scopolamine, acetylthiocholine iodide, butyrylthiocholine iodide, recombinant human acetylcholinesterase, butyrylcholiesterase from human plasma, and 5,5′–dithiobis–(2–nitrobenzoic) acid were purchased from Sigma–Aldrich (St. Louis, MO, USA). All other chemicals and solvents were of biochemical grade.

#### 3.2.2. Cholinesterases Inhibition Assay

##### In Vitro Cholinesterase Inhibition

Inhibitory activities of the tested compounds against AChE and BChE were measured according to Ellman et al. [32]. Enzyme-catalysed hydrolysis of ATCHh was carried out at 25 °C in 0.1 M phosphate buffer (pH 8.0) containing 0.25 U AChE, 1 mM ATCHh as substrates, and 0.1 mM of DTNB. In experiments with BuChE, 0.1 M phosphate buffer (pH 7.0) and BuTChh as a substrate were used. The tested compounds were pre-incubated with the enzyme for two minutes prior to the addition of the substrate and the subsequent recording of hydrolysis kinetics. The rate of substrate hydrolysis was measured for two minutes by a Shimadzu UV-1800 spectrophotometer (Shimadzu Co., Kyoto, Japan) at a wavelength of 412 nm. A sample without an inhibitor was used as a control (100% cholinesterase activity). A sample without a substrate was used as a blank. Experiments were performed in triplicate. IC_50_ values (inhibitor concentration required to inhibit enzyme activity by 50%) were determined using Origin 8.5 from the Hill equation (1):(1)$$\frac{E}{E_{max}}=\frac{{\left[I\right]}^{n}}{{IC}_{50}^{n}+{\left[I\right]}^{n}}$$
where *E*—enzyme activity in the presence of the tested compound; [*I*]—compound concentration.

The inhibition constants (*K*_i_) for human AChE and BuChE were determined by Ellman’s method [32] at 412 nm and 25 °C in 0.1 M sodium phosphate buffer (pH = 8.0 for AChE and pH = 7.0 for BuChE) with ATCHh or BuTCh concentrations in the range of 0.1–1.0 mM. A UV-1800 Shimadzu (Shimadzu Co., Kyoto, Japan) spectrophotometer was utilised to conduct the measurements. The final enzyme concentration in the assays was in the picomolar range. In order to ascertain the levels of spontaneous hydrolysis of the enzyme and substrate in the absence of an inhibitor, controls were performed for each substrate concentration. The change in absorption was then recorded over a period of three minutes. The inhibition constants were determined from a Dixon plot and Cornish–Bowden transformation [33]. 

##### In Vivo Cholinesterase Inhibition

In vivo experiments involving animals were carried out in accordance with the Directive of the Council of the European Union 2010/63/EU. The protocol of experiments was approved by the Animal Care and Use Committee of the FRC Kazan Scientific Center of RAS (protocol No. 2 from 9 June 2024). CD-1 mice (weighing 25–30 g, 6 weeks old) were purchased from the Laboratory Animal Breeding Facility (Branch of Shemyakin–Ovchinnikov Institute of Bioorganic Chemistry, Puschino, Moscow Region, Russia) and were allowed to acclimate to their environment in vivarium for at least 1 week prior to the experiments. Animals were kept in sawdust-lined plastic cages in a well-ventilated room at 20–22 °C in a 12 h light/dark cycle, with 60–70% relative humidity, and given ad libitum access to a standard diet and water. Experiments were carried out between 8 a.m. and 12 a.m.

Toxicological experiments were performed using a single i.p. injection of the tested compounds in CD-1 mice weighing 20–25 g. Stock solutions of the tested compounds were diluted in ethanol or water. The final ethanol dose was 100 mg/kg. Nine different doses of the tested compounds were used with 12 mice per dose. Animals were observed 7 days after injection, and symptoms of toxicity were recorded. The LD_50_ dose causing lethal effects in 50% of the animals was taken as a criterion of toxicity. LD_50_ was determined by the method of Weiss with 95% confidence limits [34].

For the brain AChE inhibition assay, the brains of mice were removed 30 min after i.p. injection of the tested compound (experimental group, n = 6 mice) or following i.p. vehicle injection (control group, n = 6 mice). The brains were subsequently frozen in liquid nitrogen. Whole-brain homogenates were prepared in a Potter homogeniser with 0.05 M Tris–HCl, 1% Triton X–100, 1M NaCl, and 2 mM ethylenediaminetetraacetic acid (pH 7.0; 4 °C; 1 part brain to 2 parts of buffer). The homogenate was centrifuged (14,000× *g* rpm; t = 4 °C) for 10 min using an Eppendorf 5430R centrifuge with an FA–45–30–11 rotor (Eppendorf AG, Hamburg, Germany). AChE activity in brain homogenates was measured using the method described by Dingova et al. [35] with modification. Briefly, a 50 µL aliquot of supernatant was incubated for 30 min with 5 µL of 0.5 mM tetra–isopropyl pyrophosphoramide (iso–OMPA) as a specific BChE inhibitor. Next, the enzyme-catalysed reaction was initiated by adding 10 µL of ATCH (final concentration 1.5 mM) as a substrate. Following 10, 20, or 30 min of incubation with the substrate at 25 °C, the reaction was stopped by adding neostigmine (final concentration 0.01 M). After diluting samples 25 times in a 50 mM phosphate buffer (pH 8.0), DTNB (0.1 mM) was added. The production of yellow 5–thio–2–nitro–benzoate anion resulting from the reduction of DTNB by thiocholine (the product of enzymatic hydrolysis of ATCH) was measured spectrophotometrically using a Shimadzu UV-1800 spectrophotometer (Shimadzu Co., Kyoto, Japan) according to Ellman’s method [32]. The rate of thiocholine production over 20 min (from 10 to 30 min) was calculated. Brain samples of the control group were used as a control (100% of cholinesterase activity). The sample without substrate was used as a blank. All measurements for each brain sample were performed in triplicate. AChE activity was expressed in relation to the amount of total protein, which was determined by the Bradford method [36].

#### 3.2.3. Protection Against POX Toxicity

For protection of mice against 2 × LD_50_ POX, the experimental scheme developed by Lenina et al. was used [37]. Briefly, for the POX toxicity shift assay, atropine (15 mg/kg, i.p) was administered 1 min after i.p. injection of 2 × LD_50_ POX. The tested compound was administered via i.p. before POX injection. The ratio of the number of mice that did not survive after challenge with 2 × LD_50_ POX to the total number of challenged mice was used as a criterion of toxicity shift.

#### 3.2.4. Novel Object Recognition Test

The experiments were carried out on CD1 mice of both sexes weighing 24–25 g. The novel object recognition test was conducted over a period of three days [29]. On the first day, the mice were placed individually into the square testing arena with black walls (50 cm in length, 50 cm in width, 38 cm in height) for 5 min without any objects. On the second day, two identical objects were placed in the central part of the arena, and the mice were allowed to explore the objects for 10 min. Test compounds were administered prior to testing on the second day. The induction of amnesia was facilitated by i.p. injection of scopolamine (1.5 mg/kg) 50 min prior to testing. Tested compounds were administered i.p. 30 min prior to testing. The control group of animals was administered with an equivalent amount of vehicle. On the third day, the mice were presented with a familiar and a novel object for 10 min. The time of exploration of each object by the mice was recorded using a digital camera. After each test, the arena was cleaned with a solution of 70% ethanol. At the end of the test, the preference index (exploration of novel object/total exploration time × 100) was calculated.

#### 3.2.5. Effect on Motor Function

To analyse motor function, rotarod and open-field tests were carried out. Mice were randomly divided into 3 groups (n = 12): a control group of mice and groups of mice treated with tested compounds. Tests were conducted 1 h after compound exposure. To assess motor function, mice were first placed on a rotarod treadmill (Ugo Basile, Italy) [38]. Rotation speed was fixed at 15 rpm, and the time that each mouse remained on the rod was recorded as the score, with a maximum duration of 300 s per trial. Each mouse underwent 3 trials. After recording the length of time that mice managed to remain on the rotarod (latency to fall), the average of three trials was used for further analysis. Following the rotarod test, motor function was studied in an open field apparatus (Open Science, Krasnogorsk, Russia), as described in [39]. The numbers of rearing and head dips were estimated as indexes of exploratory activity. To assess motor function, the distance travelled in centimetres in the arena was measured, using VideoMot2 software (Version 8.02 (131) 03-07-17 (TSE Systems, Bad Homburg, Germany).

#### 3.2.6. Statistics

All data processing was performed using OriginPro 8.5. The results are expressed as the mean ± standard deviation. ANOVA statistics with Tukey’s post hoc test were used to analyse the results of the novel object recognition test. The Mann–Whitney test was used to analyse the locomotor activity. Significance was tested at the 0.05 level of probability (p). Relative risk of death was investigated using Cox analysis.

### 3.3. Molecular Modelling

#### 3.3.1. Molecular Docking

The crystal structure of human butyrylcholinesterase (hBuChE) complexed with the ligand (PDB ID: 6ESY) was selected as the template, while the crystal structure of human acetylcholinesterase (hAChE) (PDB ID: 7E3H) was used for comparison. Protein alignment and superposition were performed using the Protein Align/Superpose function in the Molecular Operating Environment (MOE) to evaluate structural overlap and similarity. This analysis facilitated the investigation of the active site architecture and binding pockets of both proteins.

#### 3.3.2. Molecular Dynamics Simulation

Molecular dynamics (MD) simulations were performed using the Desmond simulation package (Schrödinger Release 13.5). The simulation systems were prepared by embedding the structures of compounds **2b** and **3c** (based on PDB IDs: 7E3H and 6ESY, respectively) into a cubic simulation box with a 10 Å buffer distance to establish a hydrated environment. Periodic boundary conditions were applied, and the box edge length was adjusted to accommodate the protein–ligand complex plus buffer. To neutralise the system and achieve physiological ionic strength (150 mM), Na^+^ and Cl^−^ ions were added using the Autoionize tool in Desmond, with a tolerance of ±0.1 charges to ensure neutrality. The systems were parameterised using the OPLS4 force field, which is a widely used force field for simulating biomolecular systems. This force field provides a balanced description of protein, lipid, nucleic acid, and small-molecule interactions. The SPC (Simple Point Charge) water model was used to represent the solvent, as it is a commonly used model that provides a good balance between computational efficiency and accuracy.

The NPT ensemble in the Desmond package was applied for system minimisation and equilibration. MD simulations were conducted for 100 ns for both compounds, with trajectory data recorded every 200 ps. Throughout the simulations, the temperature and pressure were maintained at 300 K and 1.01325 bar, respectively. Post-simulation analyses were conducted using the Simulation Interaction Diagram tool in Desmond to evaluate ligand–protein interactions. The stability of the MD simulations was assessed by monitoring the root mean square deviation (RMSD) of both ligand and protein atom positions over time. Convergence criteria were based on the RMSD values reaching a plateau, indicating the system had achieved thermodynamic equilibrium, with minimal structural drift and consistent sampling of the conformational space.

#### 3.3.3. ADMET Predictions

The Wenxin bio-computing large model developed by Baidu and the Baidu intelligent cloud high-performance computing cluster were used to build a bio-computing high-performance computing platform PaddleHelix to predict the ADMET properties of compounds (https://paddlehelix.baidu.com/news/achieve, accessed on 8 April 2025). The ADMET prediction model is based on Baidu deep learning technology, predicting the basic physical and chemical properties of a given compound, ADMET properties, biochemistry, and another 50+ indicators. The prediction process was performed using the default settings.

## 4. Conclusions

In summary, bisuracils **2b** and **3c** exhibit potent inhibitory effects and exceptional selectivity for AChE over BuChE. Bisuracil **2b** demonstrated significant prophylactic potential against OP poisoning by protecting AChE from irreversible inhibition by POX, while bisuracil **3c** effectively inhibited brain AChE activity in vivo to improve memory deficits in a scopolamine-induced amnesia model in mice. Importantly, neither compound showed adverse effects on locomotor or exploratory activity, as confirmed by behavioural tests. These findings highlight the potential of bisuracils **2b** and **3c** as promising candidates for further development in the treatment of neurodegenerative disorders and as protective agents against OP poisoning.

## Data Availability

Data are contained within the article or the Appendix A.

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
