# Peer review of "Ionic and Non-Ionic Counterparts Based on Bis(Uracilyl)Alkane Moiety with Highest Selectivity Towards Acetylcholinesterase for Protection Against Organophosphate Poisoning and Treating Alzheimer’s Disease"

_ijms, 2025, doi:10.3390/ijms26083759_

Round 1
Reviewer 1 Report
Comments and Suggestions for Authors
- Clarify the abstract by avoiding run-on sentences. Break up sentence 14 -17 into two parts for clarity.
- Sentence line17 - 20 is not clear...e.g., These bisuracils were which.... makes the sentence less clear.
- Revisit sentence 25 - 26...eg., 3C that ...is not clear.
- In your introduction section, please provide the explanation/mechanism for how reversible ChEIs rescue OPs toxicity... Is it competitive antagonism? And how effective would this be in case of OPs "aging" effect?
- Please revise sentence 59....current AD pharmacotherapy is not solely based on the cholinergic hypothesis.
- Identify the lead compound alluded to on line 75 of the introduction.
- Perhaps, add a table of predicted ADMET for these compounds.
Review the manuscript for English grammar.
Reviewer 2 Report
Comments and Suggestions for Authors
Dear Editor,
Thank you for the opportunity to review this manuscript. I sincerely appreciate the trust you have placed in me. Please find my review comments below.
The submitted manuscript entitled “Ionic and Non-Ionic Counterparts Based on Bis (Uracilyl) Alkane Moiety with Highest Selectivity Towards Acetylcholinesterase for Protection Against Organophosphate Poisoning and Treating Alzheimer’s Disease” seems potential publication in this journal. The article is well-structured, and findings are effectively presented. The study provides an evaluation of ionic and non-ionic bisuracil-based acetylcholinesterase (AChE) inhibitors. The ionic bisuracils demonstrated exceptional in vitro selectivity for AChE over butyrylcholinesterase (BuChE), while non-ionic counterparts showed blood-brain barrier (BBB) permeability and cognitive enhancement in vivo. The manuscript is scientifically robust, relevant, and well-executed. Minor editorial and methodological clarifications will further improve its quality and readability
Address the following comments:
- Consider simplifying the abstract, currently its dense and too detailed.
- Docking and MD simulations are included as part of computational study. However, more detail on software parameters, force fields, and convergence criteria would enhance reproducibility.
- The discussion section could benefit from a deeper comparison with currently approved drugs (e.g: rivastigmine, galantamine), especially in pharmacokinetic or BBB permeability terms.
- Since mixed-type inhibition is discussed, consider including more mechanistic detail(e.g: binding mode correlation with inhibition kinetics).
- Some figure references (e.g: Figures 3–10) lack clear placement or visibility in the text. Ensure all figures are appropriately placed and labeled.
- How Consider standardization of chemical naming and units throughout manuscript for the better consistency.

Reviewer 3 Report
Comments and Suggestions for Authors
Summary:
The article describes de process of novel compounds made by the group with the capacity to bind
Acetylcholinesterase and Butyl cholinesterase, based on a previous compound made by the group. The
focus was to make ionic and non-ionic variants of these compound to improve the potential of using
these compounds in Organophosphates poisoning (OP) or with potentian to be used in neurogenerative
disease. The synthetized compounds were evaluated by MALDI-TOF and inhibiton curves the determine
inhibition capacity of the compounds to the AChe and BuChe. They also made in vivo experiments to
determine the lethal dose of the compounds, especially the variants 2b (ionic) and 3c (non-ionic). They
evaluated the potential to use the compound to pre-treat OP poisoning with compound 2b and more
importantly they evaluated behavior changes with compound 3c with novel object test and motor
capacity showing potential of these compound to be used in further neurodegenerative disease studies
Comments
-The article is well written and follows comprehensive narrative order.
-In introduction they highling the Ache receptors in Alzheimer's and neurogenerative desease as the
primary factor and only two cites are forwarded in these affirmation and the reality is that the literature
is more nuanced. I suggest that they add more article and represent this nuance in the introduction.
